# Multi-Person 2-D Positioning Method Based on 77 GHz FMCW Radar

**DOI:** 10.3390/mi14061246

**Published:** 2023-06-14

**Authors:** Xiaohong Huang, Zedong Ju, Jikun Zhu, Rundong Zhang

**Affiliations:** 1College of Artificial Intelligence, North China University of Science and Technology, Tangshan 063210, China; tshxh@163.com (X.H.); jzd2015ss@163.com (Z.J.); meredithzjk@163.com (J.Z.); 2Hebei Key Laboratory of Industrial Intelligent Perception, Tangshan 063210, China; 3College of Management, North China University of Science and Technology, Tangshan 063210, China

**Keywords:** multi-person 2-D positioning, 77 GHz FMCW radar, beam scanning, multi-channel respiratory spectrum superposition, target center selection

## Abstract

As the world’s population ages, technologies that enable long-term non-contact monitoring of patients are of great research significance. For this purpose, we propose a multi-person 2-D positioning method based on a 77 GHz FMCW radar. In this method, we first perform beam scanning processing on the data cube acquired by the radar and obtain the distance–Doppler–angle data cube. Then, we eliminate interfering targets through a multi-channel respiratory spectrum superposition algorithm. Finally, we obtain the distance and angle information of the target by the target center selection method. The experimental results show that the proposed method can detect the distance and angle information of multiple people.

## 1. Introduction

The growing number of elderly individuals has resulted in a surge of age-related illnesses, including but not limited to hypertension, coronary heart disease, chronic bronchitis, pneumonia, arthritis, osteoporosis, and dementia [1]. The above situation has led to an increasing demand for medical services and a shortage of medical professionals [2,3,4]. For this reason, it is necessary to develop long-term non-contact monitoring technology for special populations. Currently, the field of non-contact monitoring primarily utilizes infrared sensors, video sensors, and radar sensors. Among them, the infrared sensor is susceptible to temperature changes, which can impact the accuracy of detection results [5]; the visual sensor lacks the capability to penetrate obstacles, and its use may pose a risk of infringing on the privacy of individuals [6]; and the radar sensor is capable of analyzing echo data to calculate the distance, angle, and speed of the target. Unlike other sensors, it is not easily affected by external factors such as temperature, light or clothing [7,8,9]. Based on the above characteristics, the use of radar-based human positioning technology has garnered the interest of numerous scholars.

In the early days, continuous wave (CW) radar was applied in the field of human position detection [10,11,12,13]. However, the above method cannot detect multi-person location information. To address such limitations, single-input multiple-output (SIMO) and multiple-input multiple-output (MIMO)-type radars were introduced [14,15,16]. The above method can provide distance and angle information on multiple targets but cannot estimate the number of targets well.

To this end, Jana et al. allowed multi-person tracking with a single UWB radar equipped with the minimal antenna array needed for trilateration [17]. Although this method reduces the complexity of data processing, it also reduces detection accuracy. Bo G et al. correctly combined observations from multiple radar nodes using a probabilistic data fusion framework [18]. Based on the measurement principle of mechanically scanned frequency modulated continuous wave (FMCW) radar, Zhang et al. proposed a scanning denoising (SD) method and an improved normal distribution transform (NDT) algorithm for radar ranging and positioning was proposed [19]. Although the above method can detect the target location information, there are still the following problems: multipath interference problem; effective spectrum energy is low; and inaccurate selection of target center position.

FMCE radar, due to its high carrier frequency and large bandwidth, is capable of forming a multi-transmit and multi-receive antenna array. This makes it a popular choice for object perception, identification, and positioning applications. Therefore, this paper chooses a 77 GHz FMCW radar platform for research.

To solve the above problems, this paper proposes a 2-D positioning method for multiple targets based on a 77GHz FMCW radar. Among them, for the problems of multipath interference and insufficient effective spectrum energy, this paper proposes a multi-channel efficient spectrum superposition algorithm. In addition, this paper proposes a target center selection method.

## 2. Date Cube Generation

Figure 1 shows the flow of the data cube generation using FMCW radar. When the human body breathes normally, the main movement of the thoracic surface is breathing. Therefore, the body surface movement of the chest can be expressed as
(1)$$x(t)=a_{r}\sin(2\pi f_{r}t)+n(t)$$
where $a_{r}$ is the chest amplitude due to respiration, $f_{r}$ is the respiration rate, and n(t) is the noise signal. The transmitted signal can be expressed as
(2)$$Y(t)=\cos(2\pi f_{c}t+\pi \frac{B}{T_{c}}t^{2})$$
where $f_{c}$ is the carrier frequency, B is the total bandwidth, and $T_{c}$ is the duration. Assume the initial distance between the radar and the body is $d_{0}$. Then, the received signal can be expressed as
(3)$$R(t)=\cos\left\{2\pi f_{c}\left[t-\frac{2\left[d_{0}+x(t)\right]}{c}\right]+\pi \frac{B}{T_{c}}{\left[t-\frac{2\left[d_{0}+x(t)\right]}{c}\right]}^{2}\right\}$$
where c is the speed of light. Then, the intermediate frequency (IF) signal can be expressed as
(4)$$B(t)=\exp\left\{j\left[2\pi \frac{2Bd_{0}}{cT_{c}}t+\frac{4\pi \left[d_{0}+x(t)\right]}{\lambda }\right]\right\}$$
where λ is the maximum wavelength of the signal. The sampled signal can be expressed as
(5)$$B(mT+nT_{s})=\exp\left\{j\left[2\pi \frac{2Bd_{0}}{cT_{C}}nT_{s}+\frac{4\pi \left[d_{0}+x(mT)\right]}{\lambda }\right]\right\}$$
where m and n represent the index number of chirps and the index number of the sampling point in each chirp, respectively; T is the slow time sampling period, and $T_{s}$ is the fast time sampling period.

Assume there are X targets in the space, and the initial distances from the radar are $d_{1}$, $d_{1}$, $d_{2}$, ⋯, $d_{X-1}$, $d_{X}$, respectively. After introducing the multi-receiving antenna model, Formula (5) can be rewritten as
(6)$$B\left(mT+nT_{s},i\right)=\sum_{k=1}^{X}\exp\left\{j\left[2\pi \frac{2Bd_{k}}{cT_{c}}nT_{s}+\frac{4\pi \left[d_{k}+x_{k}\left(mT\right)\right]}{\lambda }+\frac{2\pi \left(i-1\right)l}{\lambda }\sin\theta_{k}\right]\right\}$$
where l is the distance between adjacent receiving antennas and $\theta_{k}$ represents the angle between the target k and the radar.

To obtain the distance–Doppler–space dimension data cube, this paper performed fast-time dimension FFT and slow-time dimension FFT on $B\left(mT+nT_{s},i\right)$. It is worth noting that in order to reduce spectrum leakage, this paper added the Hanning window to the IF signal before performing fast-time dimensional FFT processing [20].

## 3. Methods

Figure 2 shows the block diagram of the multi-person 2-D positioning method, which is divided into three parts (A: beam scanning; B: multi-channel efficient spectrum superposition algorithm; C: target center selection method). The corresponding functions of A, B, and C are as follows: initial screening of target locations; eliminate static and dynamic clutter; pick the center point of the target.

### 3.1. Beam Scanning

Digital beam forming (DBF) can realize the in-phase superposition of multiple channel signals [21], and its process is shown in Figure 3. After the range–Doppler–space dimension data cube is processed by DBF, it can be expressed as
(7)$${\left[Y_{1}^{\prime }\cdots Y_{x}^{\prime }\cdots Y_{X}^{\prime }\right]}^{T}={\left[W_{1},W_{2},\cdots ,W_{k},\cdots ,W_{K}\right]}^{H}{\left[Y_{1}\cdots Y_{x}\cdots Y_{X}\right]}^{T}$$
where $W_{x}$ is the weight matrix, ${\left[\right]}^{H}$ represents the transpose of matrix [], $Y_{x}$ indicates that each antenna extracts information of different targets, ${\left[\right]}^{T}$ denotes transposition of the matrix []. Among them, $W_{x}$ can be expressed as:(8)$$W_{x}={\left[1,e^{j\frac{2\pi d}{\lambda }\sin\theta_{x}},\cdots ,e^{j\frac{2\pi (i-1)l}{\lambda }\sin\theta_{x}}\right]}^{T}$$

After the above processing, the distance–angle–Doppler dimension data cube is obtained, and expressed as
(9)$$M\left(R,D,A\right)$$

### 3.2. Multi-Channel Efficient Spectrum Superposition Algorithm

The breathing amplitude is much larger than the heartbeat amplitude, so this paper judges the 2-D position of the human body based on breathing. The range of speeds at which breathing causes movement of the chest is as follows:(10)$$v_{h-\min}=2a_{h}f_{h-\min}$$
(11)$$v_{h-\max}=2a_{h}f_{h-\max}$$
where $f_{h-\min}$ and $f_{h-\max}$ are the minimum and maximum values of the normal respiratory rate of the human body, respectively. Assume that the total number of transmitted pulses is M. Then, the velocity resolution can be expressed as
(12)$$v_{\Delta }=\frac{\lambda }{2MT}$$

Combining the above formulas, the effective range of respiration in the Doppler dimension can be obtained as
(13)$$\left[P_{1}-P_{2}\right]$$
where $P_{1}$ and $P_{2}$ are the lower and upper bounds, respectively. Among them, $P_{1}$ and $P_{2}$ can be expressed as
(14)$$P_{1}=\frac{4a_{h}f_{h-\min}MT}{\lambda }$$
(15)$$P_{2}=\frac{4a_{h}f_{h-\max}MT}{\lambda }$$

Accumulate the energy of Doppler slices between Doppler dimensions $\left[P_{1}-P_{2}\right]$, where the energy value of each point is
(16)$$E\left(R,A\right)=\sum_{p=p_{1}}^{p_{2}}M_{R,P,A}\left(R,P,A\right)$$where $M_{R,P,A}$ is the distance, Doppler slice, and angle dimension data cube. At this point, the processed data matrix is expressed as
(17)$$Y_{\left(R,A\right)}$$

### 3.3. Target Center Selection Method

When the FMCW radar system is used to locate multiple 2-D targets, if the weak target reference module has one or more strong targets, occlusion is likely to occur. In order to solve the above problems, the data cube $Y_{\left(R,A\right)}$ has carried out different CFAR detection in the distance dimension and angle dimension, respectively. Figure 4 shows the CFAR detection flow chart. When detecting, multiple targets may be close in distance. Therefore, in order to reduce the problem of target occlusion, we use SO-CFAR in the distance dimension. At this point, μ can be expressed as
(18)$$\mu =\min(\mu_{1},\mu_{2})$$

In order to solve the edge clutter problem, and this article has a high angular resolution, the angular dimension uses GO-CFAR. At this point, μ can be expressed as
(19)$$\mu =\max(\mu_{1},\mu_{2})$$

At this point, $Y_{\left(R,A\right)}$ is denoted as $Y_{\left(R,A\right)}^{\prime }$ after two CFAR tests. Finally, the center points of multiple targets are selected, and the process is as follows:The area range of detected X targets can be expressed as
(20)$$Y^{1-X}=\left[Y_{\left(r_{1},a_{1}\right)}^{1},Y_{\left(r_{2},a_{2}\right)}^{2},\cdots ,Y_{\left(r_{x},a_{x}\right)}^{x},\cdots ,Y_{\left(r_{X},a_{X}\right)}^{X}\right]$$Select the maximum energy point in the area $Y_{\left(r_{x},a_{x}\right)}^{x}$, and record its distance and angle information.Record the location information of X targets separately, and express it as
(21)$$O^{X}=\left[\left(r_{1},a_{1}\right),\left(r_{2},a_{2}\right),\cdots ,\left(r_{x},a_{x}\right),\cdots ,\left(r_{X},a_{X}\right)\right]$$

After the above processing, the distance and angle information of multiple targets can be obtained.

## 4. Results and Discussion

We conducted the experimental validation using the commercial Texas Instruments IWR1443BOOST mmWave radar sensor (Texas Instruments, State of Texas, America), which can form an FMCW radar in a MIMO system. The system parameters were $f_{c}=77\;\mathrm{GHz}$; B=3.90 GHz; $T_{c}=52\;\mu s$; λ=3.9 mm; N=512; $T_{s}=0.07\;\mu s$; M=256; T=50 ms.

The calculation formulas of distance resolution and angle resolution are
(22)$$\Delta R=\frac{c}{2B}$$
(23)$$\Delta A=\frac{2}{I}$$

Under the system parameters, the distance and angle resolutions are 4 cm and 15°, respectively.

Figure 5 shows the experimental scene. Two young people stood on the left and right halves of the radar, and there were two anti-angles (strong interference). We measured the distance with a tape measure and calculated the angle (expected results). We collected data for 12.8 s each time, and implemented the method in MATLAB (MagicBook 16 Pro, ordinary configuration of computer, the cost is low). To quantify the performance of the method, the error and average error were introduced into the performance analysis, which are expressed as
(24)$$ME=HR_{mea-l}-HR_{rea-l}$$
(25)$$AE=\left[\sum_{l=1}^{L}\left(HR_{mea-l}-HR_{rea-l}\right)\right]/L$$
where L is the number of measurement groups, $HR_{mea-l}$ is the measured value, and $HR_{rea-l}$ is the expected results. It should be noted that the clockwise is positive.

Figure 6 shows a set of experimental results (target A, target B). Among them, the target to be measured belongs to the far-field range. Figure 6a shows the results of the beam scanning, and the distance and angle information of the target cannot be resolved. Figure 6b shows the imaging results after multi-channel effective spectrum superposition. It can be seen from the figure that the positions of the two targets are obvious, and their positions are $\left(2\;m-2.5\;m\right)$, $\left(-50^{\circ },-20^{\circ }\right)$; $\left(2\;m-2.5\;m\right)$, $\left(10^{\circ },40^{\circ }\right)$. Figure 6c shows the results after 2-D CFAR detection. Figure 6c shows the clustered results. It can be seen from the figure that the dynamic clutter is effectively eliminated, and the two targets are very easy to distinguish. Finally, we used the target center selection method to obtain the positions of the two targets: 2.22 m, $-34^{\circ }$; 2.34 m, $24^{\circ }$ (expected results: 2.29 m, $-32^{\circ }$; 2.29 m, $32^{\circ }$). Calculated according to the formula, we know that the error is 0.07 m, $2^{\circ }$; 0.05 m, $8^{\circ }$.

We validated the proposed approach by conducting 15 experiments (three experiments at the same location) on two subjects, differing in height (170 cm–180 cm) and in age (22 years–26 years). We collected data for 12.8 s each time. Table 1 shows the results of the experimental validation (target A, target B). The distance and angle average errors of target A and target B are 0.07 m, $3.6^{\circ }$; 0.06 m, $4.9^{\circ }$.

Figure 7 shows a set of experimental results (target A, target B, and target C). The measured values are as follows: 2.22 m, $-34^{\circ }$ (target A); 1.99 m, $-5^{\circ }$ (target B); and 2.59 m, $21^{\circ }$ (target C). The reference value is as follows: 2.29 m, $-32^{\circ }$ (target A); 1.95 m, $0^{\circ }$ (target B); and 2.30 m, $24^{\circ }$ (target C). Table 2 shows the results of the experimental validation (target A, target B, and target C). The distance and angle average errors of target A target B and target C are 0.07 m, $1.8^{\circ }$; 0.05 m, $5.3^{\circ }$; and 0.09 m, $2.3^{\circ }$. This shows that the proposed method can effectively detect multi-person 2-D information based on a 77 GHz FMCW radar. Compared with other traditional methods, this method has the following advantages: it can effectively eliminate static interference and dynamic interference and it can increase the target center point selection accuracy.

## 5. Conclusions

To improve the application of non-contact detection technology in the medical field, this paper proposes a 2-D multi-person positioning method based on a 77 GHz FMCW radar. This method eliminates the problem of multipath interference through the multi-channel respiratory spectrum superposition algorithm and accurately selects the center position of the target through the multi-target center point selection method. Through experiments, it is found that the method can effectively detect the distance and angle information of multiple people. The proposed method has important potential significance in the field of telemedicine monitoring, including human perception, recognition, and localization. This provides an important foundation for human body pose recognition and vital sign detection.

## Figures and Tables

**Figure 1 micromachines-14-01246-f001:**
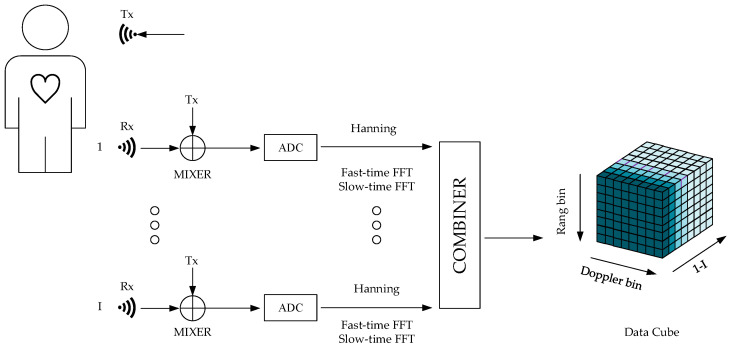
Block diagram of the data cube generation using FMCW radar. Tx: transmitting antenna, I: index number of receiving antennas, Rx: receiving antenna, MIXER: mixer, ADC: analog digital converter, FFT: fast Fourier transform.

**Figure 2 micromachines-14-01246-f002:**
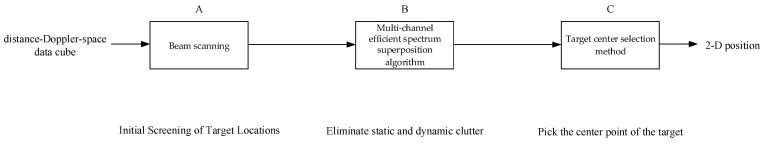
Block diagram of the multi-person 2-D positioning method.

**Figure 3 micromachines-14-01246-f003:**
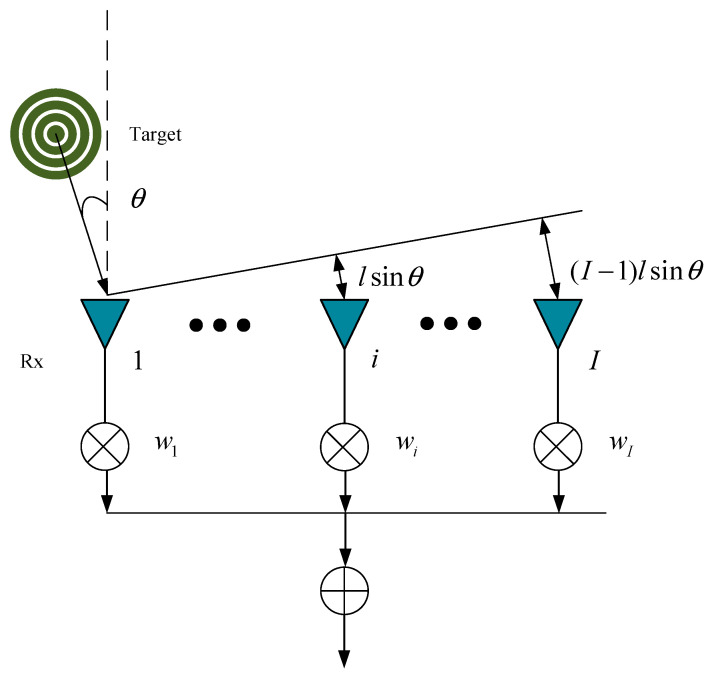
Block diagram of the DBF technology.

**Figure 4 micromachines-14-01246-f004:**
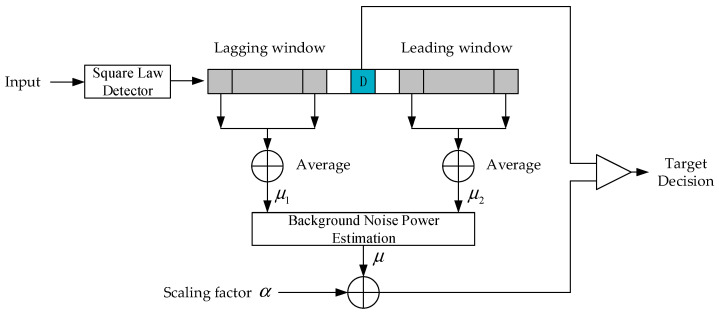
Block diagram of the CFAR detection.

**Figure 5 micromachines-14-01246-f005:**
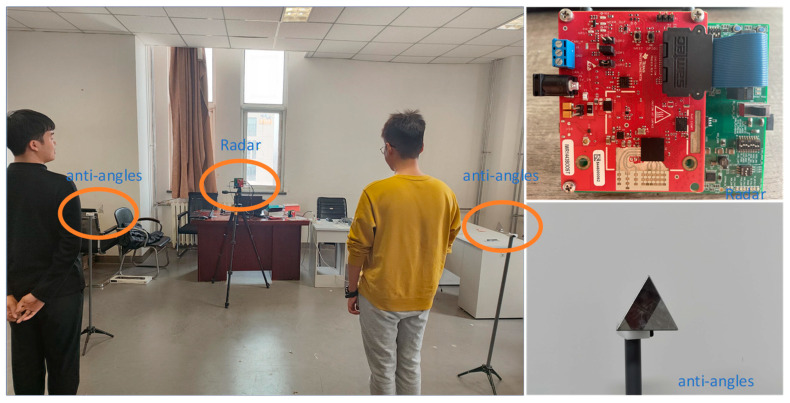
Experimental environment.

**Figure 6 micromachines-14-01246-f006:**
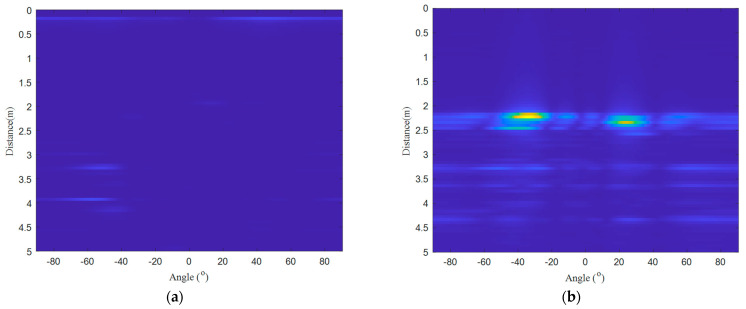
Experimental results (target A, target B): (**a**) results of the beam scanning; (**b**) imaging results after multi-channel effective spectrum superposition; (**c**) results after 2−D CFAR detection; (**d**) clustered results, target A is on the left, target B is on the right.

**Figure 7 micromachines-14-01246-f007:**
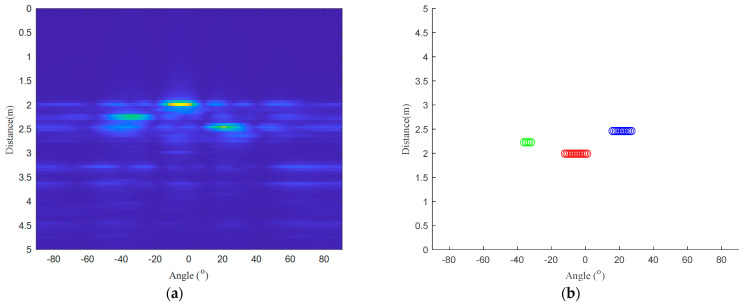
Experimental results (target A, target B, target C): (**a**) imaging results after multi-channel effective spectrum superposition; (**b**) clustered results, target A is on the left, target B is on the middle, target C is on the right.

**Table 1 micromachines-14-01246-t001:** Results of the experimental validation (target A, target B).

	Target A (Range/m; Angle/°)						Target B (Range/m; Angle/°)					
	Range $HR_{mea-l}$	Range $HR_{rea-l}$	ME	Angle $HR_{mea-l}$	$\mathrm{Angle}\;HR_{rea-l}$	ME $/{}^{\circ }$	Rang $HR_{mea-l}$	$\mathrm{Range}\;HR_{rea-l}$	ME	Angle $HR_{mea-l}$	$\mathrm{Angle}\;HR_{rea-l}$	ME
1	2.22	2.29	0.07	−34	−32	2	2.34	2.29	0.05	24	32	8
2	2.22	2.29	0.07	−34	−32	2	2.28	2.29	0.01	24	32	8
3	2.22	2.29	0.07	−34	−32	2	2.28	2.29	0.01	24	32	8
4	1.93	2.00	0.07	−17	−13	4	2.11	2.15	0.04	19	24	5
5	1.99	2.00	0.01	−17	−13	4	2.11	2.15	0.04	18	24	6
6	1.99	2.00	0.01	−17	−13	4	2.11	2.15	0.04	16	24	8
7	2.17	2.15	0.02	−28	−24	4	2.28	2.15	0.14	13	14	1
8	2.17	2.15	0.02	−28	−24	4	2.23	2.15	0.08	16	14	2
9	2.17	2.15	0.02	−29	−24	5	2.23	2.15	0.08	15	14	1
10	2.05	2.29	0.24	−29	−32	3	2.28	2.15	0.13	23	24	1
11	2.11	2.29	0.18	−27	−32	5	2.28	2.15	0.13	25	24	1
12	2.11	2.29	0.18	−28	−32	4	2.28	2.15	0.13	22	24	2
13	1.99	2.00	0.01	−17	−13	4	2.28	2.29	0.01	25	32	7
14	1.99	2.00	0.01	−16	−13	3	2.28	2.29	0.01	24	32	8
15	1.99	2.00	0.01	−17	−13	4	2.34	2.29	0.05	24	32	8
	AE/**m**		0.07	AE/${}^{\circ }$		3.6	AE/**m**		0.06	AE/${}^{\circ }$		4.9

**Table 2 micromachines-14-01246-t002:** Results of the experimental validation (target A, target B, target C).

Number	Target A (m; °)		Target B (m; °)		Target C (m; °)	
	Range $HR_{mea-l}$	Angle $HR_{mea-l}$	Range $HR_{mea-l}$	Angle $HR_{mea-l}$	Range $HR_{mea-l}$	Angle $HR_{mea-l}$
1	2.22	−32	1.99	−5	2.59	24
2	2.28	−33	2.05	−6	2.46	22
3	2.28	−35	1.99	−6	2.46	22
4	2.23	−34	0.94	−5	2.46	22
5	2.17	−35	1.00	−6	2.46	23
6	2.17	−34	0.94	−4	2.52	24
AE/**m**	0.07	1.8	0.05	5.3	0.09	2.3

## Data Availability

The data that support the plots within this paper are available from the second author upon reasonable request.

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
