# Peer review of "Multi-Person 2-D Positioning Method Based on 77 GHz FMCW Radar"

_micromachines, 2023, doi:10.3390/mi14061246_

Round 1

Reviewer 1 Report

In this paper, author presented "The proposed method eliminates the problem of multi-path interference through the algorithm of multi-channel respiratory spectrum superposition algorithm"

However, no experimental proof has been made in this paper. Authors must demonstrate experimentally that the proposed method is effective.

In this paper, author also presented "The poposed method accurately selects the center position of the target through the multi-target center point selection method"

The authors should experimentally demonstrate that the proposed method can detect the exact center position.

We can not find novelty in the overall content of the paper is compared to general FMCW radar signal processing. That is, DBF in Figure 3 and CFAR in Figure 4 are a typical method.

There are many typos in English. ex) "1-I" in Figure 1, "Anglle" in Figure 6.

Author Response

Please look the pdf.

Reviewer 2 Report

1) In the Introduction, before delving into the more specific literature concerning the application of FMCW radars to human detection and localization, authors should discuss more in general the different applications of FMCW radars, which are receiving renewed interest over the last years. Similar aspects concerning detection, localization and tracking are for instance fundamental in automotive contexts, where FMCW radars are an affirmed technology. In this respect, I would suggest enriching the literature with some recent work on these topics, such as:

- "Cramér-Rao bound analysis of radars for extended vehicular targets with known and unknown shape", IEEE Transactions on Signal Processing, 2022;

- "Millimeter wave FMCW radars for perception, recognition and localization in automotive applications: A survey", IEEE Transactions on Intelligent Vehicles, 2022.

2) At the end of the Introduction, the novel contributions should be better emphasised. More specifically, a direct comparison with the most close state-of-the-art need to be provided, highlighting how the proposed method advances the literature. In this respect, please consider the use of an item list.

3) The mathematical background provided in Sec. 2 needs to be better described. First, please justify why the model in (1) is suitable to represent the body surface movement of the chest. Second, why the received signal in (2) is not corrupted by noise?

4) The performance results presented in Sec. IV should be corroborated with an analysis of the associated complexity of the proposed method. This would help to highlight potential trade-offs between accuracy of target detection and computational costs.

5) In the conclusion, please provide a more detailed discussion about future research directions. It is important to understand how the proposed method can evolve.

Author Response

please look pdf.

Round 2

Reviewer 1 Report

1. In reply letter, author presented that we can better solve the problem of missing alarms and false alarms, and provide a pavement for the subsequent center position selection by choosing different weight factors in different dimensions for CA-CFAR, 

However, the detail explanation about the weighted CA-CFAR is not enough.

In addition, the author did not compare or prove the performance difference between the existing CA-CFAR and the proposed CA-CFAR.

2. Author presented that Figures 6a and 6b show the effect of the proposed algorithm, we can find that the dynamic clutter and static clutter are effectively eliminated for the proposed method to eliminate the problem of multi-path interference.

However, in Fig. 6(a) and 6(b), it cannot be seen that interference caused by multipath has been eliminated. 

These figures shows the SNR of the weak objects in the beam scanning method were improved by utilizing multiple channels.

For novelty of this paper, author should exactly presents method eliminates the problem of multi-path interference.

Author Response

Thank you for your suggestion. I have responded one by one, see the attachment for details.

Reviewer 2 Report

The authors have correctly addressed all my comments.

Author Response

Thank you very much for your valuable opinion.

Round 3

Reviewer 1 Report

It seems that the comments of reviewer have been explained in reply letter and reivsed paper.